Process-based allometry describes the influence of management on orchard tree aboveground architecture

Brym Zachary T. brymz@ufl.edu 1
Ernest S.K. Morgan 2
1 Tropical Research and Education Center, University of Florida , Homestead , FL , United States of America
2 Wildlife Ecology and Conservation Department, University of Florida , Gainesville , FL , United States of America
Huang Cho-ying
Electronic publication date: 2018 Jun 8
Publication date: 2018
Volume: 6
Electronic Location ID: e4949
Received 2018 Feb 5; Accepted 2018 May 20
Copyright: ©2018 Brym and Ernest
Copyright year: 2018
Copyright holder: Brym and Ernest
License: This is an open access article distributed under the terms of the Creative Commons Attribution License, which permits unrestricted use, distribution, reproduction and adaptation in any medium and for any purpose provided that it is properly attributed. For attribution, the original author(s), title, publication source (PeerJ) and either DOI or URL of the article must be cited.
License URL: https://creativecommons.org/licenses/by/4.0/

Keywords: Malus spp., Prunus cerasus, Flow Similarity, Canopy structure, Biomass, WBE model, Allometry, Tree architecture

Funding: Utah State University Graduate Student Senate Research Project Grant and a Utah State University Ecology Center Assistantship and Research Support Award This research was supported by a Utah State University Graduate Student Senate Research and Project Grant and a Utah State University Ecology Center Assistantship and Research Support Award. The funders had no role in study design, data collection and analysis, decision to publish, or preparation of the manuscript.

==============================
We evaluated allometric relationships in length, diameter, and mass of branches for two variably managed orchard tree species (tart cherry, Prunus cerasus; apple, Malus spp.). The empirically estimated allometric exponents (a) of the orchard trees were described in the context of two processed-based allometry models that make predictions for a: the West, Brown and Enquist fractal branching model (WBE) and the recently introduced Flow Similarity model (FS). These allometric models make predictions about relationships in plant morphology (e.g., branch mass, diameter, length, volume, surface area) based on constraints imposed on plant growth by physical and physiological processes. We compared our empirical estimates of a to the model predictions to interpret the physiological implications of pruning and management in orchard systems. Our study found strong allometric relationships among the species and individuals studied with limited agreement with the expectations of either model. The 8/3-power law prediction of the mass ∼ diameter relationship by the WBE, indicative of biomechanical limitations, was marginally supported by this study. Length-including allometric relationships deviated from predictions of both models, but shift toward the expectation of flow similarity. In this way, managed orchard trees deviated from strict adherence to the idealized expectations of the models, but still fall within the range of model expectations in many cases despite intensive management.

Introduction

The physical structure of a plant emerges from species-specific growth strategies for accessing scarce environmental resources such as light, water, and nutrients (Pacala & Tilman, 1994; Grossman & DeJong, 1995; Kobe, 2006). While species vary in their strategies for growing in resource-limited environments, even when resources are not limited, growth is constrained by physiological limits on processes such as photosynthesis and resource transport (Murneek & Logan, 1932; Niklas & Kerchner, 1984). The diversity of plant form that exists in nature reveals the many ways that plants evolved to balance trade-offs between external environmental and internal physiological limitations (Niklas, 1997). Despite the diversity of plant form, however, there are still emergent patterns in plant structure that may reflect the impact of underlying constraints on plant physiology (Price et al., 2010).

One aspect of plant form where physiological constraints on plant growth may be evident is aboveground morphology or architecture (Niklas, 2004). Aboveground morphology of plants often exhibits regular patterns referred to as allometric relationships (Huxley & Teissier, 1936; Lacointe, 2000). Allometric relationships describe how plant size (e.g., mass or diameter) relates to other dimensions of morphology such as branch length, surface area, or volume. These relationships are often highly constrained (i.e., show limited variance) and are typically well described by power law equations of the form: y=bxa

where y is the measurement of some trait of interest, x is a measure of plant size, and a and b represent values that describe the form of the relationship, the allometric exponent and multiplier, respectively. This equation is often expressed in log–log space which then expresses a as the slope of a linear relationship and b as the y-intercept.

Although powerful for generalizing patterns in plant architecture, allometries are often established for specific species of plants grown in specific conditions (Brown, 1997; Niklas, 2004). Various models exist attempting to explain why allometries often take the general form of a power law and why we often see a narrow range of allometric exponent values (a) in empirical data. The exact value and importance of a remains debated (Thompson, 1942; Enquist et al., 2007; Coomes, Lines & Allen, 2011; Price et al., 2012) as is the usefulness of an allometric approach to describing plant form, especially in applied settings (Le Roux et al., 2011). Applied models of plant architecture require detailed information for plant response to growing conditions and management (Le Roux et al., 2011; Lang & Lang, 2008). Therefore, the future improvement of morphological plant modeling requires availability of detailed information of plant growth and management linked with models that provide detailed plant information with mathematical perspective (Bucksch et al., 2017).

This study contributes detailed information about plant morphology for mature orchard fruit species and links the physiological information with two process-based plant allometry models that provide expected ranges of allometric exponent values that should be consistent across the diversity of plant species. However, orchard trees experience unique growing conditions that may be expected to deviate from the general patterns described in the process-based plant allometry models. The genetic material and management regimen of orchard trees provides a distinct manipulated environment and set of growing conditions. Clonal rootstocks that influence tree growth and partitioning are grafted to scion wood that produces favorable fruit (Robinson, 2007). The orchard environment is heavily subsidized with water and nutrients, reducing the effect of resource limitation on plant architecture. With high subsidies of water and nutrients, physiological constraints should be the primary influence on plant growth (Deng et al., 2012). However, tree architecture is also directly manipulated by pruning and training to improve light penetration, airflow, and fruit production (Lauri et al., 2011). Dormant season pruning, the destructive removal of branches in winter months, impacts the growth trajectory of trees by removing growing nodes and displacing growth hormones. Trees respond to pruning the following growing season with localized invigoration of retained buds and branches and an overall dwarfing of tree size (Ferree & Schupp, 2003). How controlled genetic material, environmental manipulation, and pruning interact to affect the overall allometry of an orchard tree is unknown.

We compare two variably managed orchard tree species using process-based allometry models as context to describe the effects of human manipulation on allometry. Allometries have been widely used in the management of orchard systems to predict carbon partitioning to fruit (Westwood & Roberts, 1970; Lacointe, 2000). However, the allometric relationships used in horticulture can be rigid and lack linkages to the underlying processes generating estimates of a. In contrast, a recently proposed process-based allometry model called Flow Similarity (FS) attempts to explicitly explain the variation in a by incorporating two fundamental physical processes constraining plant growth: hydraulics and biomechanics (Price et al., 2015). The hydraulic constraint is described as ‘flow similarity’, which is the condition where a constant flow rate and velocity of water is maintained through the plant vascular network by area-preserving branching (Shinozaki et al., 1964; McCulloh & Sperry, 2005). The biomechanical constraint is described as ‘elastic similarity’, where each branch grows to the structural limit at which if it were to grow any larger it would break under its own weight (McMahon & Kronauer, 1976; Niklas, 1992). The FS model explicitly recognizes that a tree grows dynamically in order to optimize water use while providing a sufficient structural architecture. New growth and branches at the distal end of a plant are likely to express flow similarity, while the trunk and basal structural branches that bear the majority of weight of the plant are likely to express elastic similarity. From this dynamic view of interacting physical constraints, FS predicts a range of a that falls within the bounds of hydraulic and biomechanical constraints (Table 1). The FS approach to expressing dynamic constraints between hydraulic and biomechanical limits is included as context for the interpretation of our analysis in addition to a more established model derived by West, Brown and Enquist that is built on similar processes but makes different predictions (WBE; 1997). WBE assumes that the interaction between biomechanical and hydraulic processes is fixed within and across species, but still offers a range of a predictions in some cases that recognizes the various physiological constraints between small and young growth that is constrained by water versus large and old growth that is constrained by weight (Table 1; Enquist et al., 2007).

Table 1 Predicted allometric relationships between (a) length, diameter, surface area and volume as formulated by the Flow Similarity model (FS) and (b) length, diameter, and mass as formulated by the West, Brown, Enquist model (WBE).

Y and X variables are listed in the first two columns. An expression for each relationship is in the third column, where αF is the expected exponent for the FS length to diameter prediction and where αM represents the set of expected exponents for the WBE predictions. The following columns represent the predicted exponents. For FS, the predictions are broken down into flow similarity, elastic similarity, and the change in exponent expected going from small to large plants (flow to elastic). This table is modified from Price et al. (preprint) with permission.

Y-variable	X-variable	Expression	Flow similarity	Elastic similarity	Predicted exponent	
(a) FS	
Length	Diameter	L=DFα	2	2/3	2 to 2/3	
Surface area	Volume	SA=VαF+1∕αF+2	3/4	5/8	3/4 to 5/8	
Diameter	Volume	D=V1∕αF+2	1/4	3/8	1/4 to 3/8	
Length	Volume	L=VαF∕αF+2	1/2	1/4	1/2 to 1/4	
Diameter	Surface Area	D=SA1∕αF+1	1/3	3/5	1/3 to 3/5	
Length	Surface Area	L=SAαF∕αF+1	2/3	2/5	2/3 to 2/5	
Y-variable	X-variable	Expression	Predicted Exponent	
(b) WBE	
Length	Diameter	L=DMα	>1 to 2/3	
Length	Mass	L=MMα	1/4	
Mass	Diameter	M=DMα	8/3	

The application of process-based allometry models to interpret the plant architecture of orchard trees provides the opportunity to understand how human management impacts the fundamental physiological constraints described by allometry and how these constraints on plant growth and morphology influence how managed trees respond to human manipulation. Are the allometric relationships of managed trees still consistent with expectations from process-based allometric models built to explain plant architecture of unmanaged trees? Or, does human manipulation of the natural architecture push orchard trees away from basic physical and biological constraints to exhibit forms with little comparison in unmanaged systems?

Materials & Methods

System

Two Rosaceous species from experimental orchard blocks at the Utah State University Kaysville Research Farm (2011–2013) in Davis County, Utah were used in this study: tart cherry (Prunus cerasus) and apple (Malus spp.). The production systems sampled for tart cherry and apple differ in management intensity and genetic complexity.

The sampled tart cherry orchard block was twenty-four years-old, near the end of peak production age for similar orchard systems. Individuals consisted of a clonal scion (cultivar: ‘Montmorency’) grafted on to closely related seedling rootstocks (Prunus mahaleb). These individuals are described generally by vigorous growth and wide crotch angles (57.1° ± 27.9). Fruit-bearing spurs, stubby twigs that grow off of main branches, tend to be located on the proximal two-thirds of parent branches (Maguylo, Lang & Perry, 2004). A multiple leader ‘open-vase’ canopy was developed in the first few years of growth by selecting three to five main structural branches for ideal orientation and branching angle. Following initial canopy development, individuals received relatively minor annual pruning, ∼10% total biomass, to improve light penetration, air flow, and fruit set. No pruning occurred for five years prior to the study.

The apple block was ten years-old and part of the NC-140 Regional Rootstock Research Project - 2003 ‘Golden Delicious’ Trial (Marini et al., 2014). The individual trees consisted of clonal scions (cultivar: ‘Golden Delicious’) grafted on to several clonal rootstocks (‘Budagovsky 9’, ‘Geneva®41’, ‘Geneva®210’, ‘Malling 26’, ‘Japan-Morioka 8’, ‘Pi-AU 56-83’). The ‘Golden Delicious’ cultivar is described as moderately vigorous with wide crotch angles (56.4° ± 36.6) and bears fruit on spurs or terminals of short shoots (Ferree & Schupp, 2003). Rootstocks primarily drive growth potential and disease resistance and provide a gradient in tree size, introduced from smallest to largest. Budagovsky 9 (B.9) is highly dwarfing, highly productive, and winter hardy (Budagovsky, 1974; Stehr, 2007; Hoover et al., 2011). Geneva®41 (G.41) is a dwarf rootstock with wide crotch angles that expresses good yield and fruit size, disease resistance, and winter hardiness (Robinson & Hoying, 2004; Fazio, Aldwinkle & Robinson, 2013). Geneva®210 (G.210) is a semi-dwarf with wide crotch angles that is disease resistant, free standing, precocious and productive (Fazio, Aldwinkle & Robinson, 2013). Malling 26 (M.26) is one of the most common dwarf rootstocks in commercial planting but is susceptible to disease and winter freezing (Hoover et al., 2011; Marini et al., 2009; Robinson, 2007). In the western United States it grows more like a semi-dwarf, as observed in our study as an intermediate-sized rootstock. Japan-Morioka 8 (JM.8) is reported as a dwarfing rootstock that is disease resistant; however, it also expresses a semi-dwarf size in some environments, as we observe in our study (Marini et al., 2009; Soejima et al., 2010). Pi-AU 56-83 is reported as a dwarfing rootstock from trials in Germany (Fischer, 2001), but other reports suggest it expresses as a semi-dwarf with high survival, vigor and fruit weight, but low production (Marini et al., 2009). All apple trees were trained, pruned, and managed consistently according to NC-140 protocols (http://www.nc140.org). Individuals were trellised and pruned heavily each year, ∼25% total biomass, to maintain one dominant central trunk, or single leader, and whorled terraces of lateral branches for bearing fruit.

Data collection

We sampled five tart cherry trees for a total of 449 branches and 19 apple trees for a total of 375 branches. The five tart cherries were chosen from a stratified random sample of over 300 available trees in the block while excluding senescent or diseased individuals. Six apple rootstocks were chosen from the experimental block to represent a gradient in tree size and superior survivorship. Three to four individuals of each rootstock were then chosen randomly, except for the industry standard rootstock, M.26, which only had one individual surviving. For each individual sampled, all branches were identified and measured for diameter and length. Branches were defined as a continuous stem between two branching nodes, while twigs were defined as stems supporting only buds or short fruit-bearing stems less than two centimeters in diameter. Branches and twigs were removed from the tree and dried in a large oven for a minimum of one week at 65 °C and weighed for biomass. Twig length and biomass measurements were limited to one random individual for cherry and each apple rootstock. Sampling occurred at least one full growing season following the last pruning event.

Branch classification

Branch morphology was classified in three ways to explore allometric patterns below the individual-level: segment, path and subtree (Fig. 1). Segment values are the data gathered directly for each branch. Segment length, for instance, is the distance between the proximal end of the branch at one branching node and its distal end at the next branching node. Path values are the data gathered for a given branch and the longest continuous length of distil branches. Path values are the unit of branch length most closely associated with the predictions of WBE (Smith et al., 2013). Subtree values are the diameter of a given branch and the total length or mass of that branch and all distil branches. Subtree values are a unit of branch classification associated with FS. The multi-dimensional morphological characteristics, surface area (π * diameter * length) and volume (π * (diameter/2)2 * length), are calculated at a segment level with path and subtree level values generated as the appropriate sum of segment level calculations.

Figure 1 An illustration of the branch-level classifications: segment, path and subtree.

The allometric analysis was conducted at segment, path and subtree branch classifications for each relationship.

Data analysis

To estimate a, log–log transformed linear relationships for combinations of morphological characteristics were evaluated using reduced major axis regression (Warton et al., 2006). For each pairwise relationship and branch classification, an estimated a with 95% confidence intervals and an r2 value were determined using the ‘SMATR’ package in R (R Core Team, 2014). Estimates of a were evaluated by aggregating all branches at the individual and species level.

Because these relationships are not always linear on log–log plots, we also examined whether polynomial fits to the data performed better (Niklas & Hammond, 2014). Polynomial fits were tested against linear fits by comparing the AICc values, but did not strongly alter the analysis. Only results of the linear models are reported in the paper.

The addition of twig lengths and mass to branch-level calculations were evaluated among the subset of individuals with the extra sampling effort, with only minor shifts in estimated a. Results of this supplemental analysis are not presented in the paper.

Data, programming code, and supplemental information for this analysis can be found freely available online at https://github.com/weecology/branch-arch/tree/master/GeneralAllometry.

Results

Allometries

Branch-level allometries for all branches collected for each species are reported for the length ∼ diameter and mass ∼ diameter relationships (Table 2). The allometries represent empirical estimates of plant architecture for mature cherry and apple trees. All but five estimated 95% confidence intervals of estimated a at the species level for path and subtree branch classifications overlap and are therefore interpreted as statistically indistinguishable (Table 2; Fig. 2). Apple and cherry trees differ in their estimated a at the path level for length ∼ volume (Fig. 2D), length ∼ area (Fig. 2F) and mass ∼ volume (Fig. 2I). Apple and cherry trees differ in their estimated a at the subtree level for surface area ∼ volume (Fig. 2B) and mass ∼ volume (Fig. 2I).

Table 2 The length ∼ diameter and mass ∼ diameter branch level allometries for all branches collected for: (a) five 24-year-old tart cherry, (b) nineteen 10-year-old ‘Golden Delicious’ apple.

The values reported are the empirically estimated allometric exponent (a) and multiplier (b) or slope and intercept in log–log space, respectively, 95% confidence intervals (CI), and r2 for each branch classification (segment, path, and subtree).

		Estimated a	95% CI	Estimated b	95% CI	r2	
(a) Cherry	
Length ∼ Diameter	Segment	0.83	0.76–0.91	0.51	0.40–0.62	0.006	
	Path	0.99	0.93–1.05	0.64	0.56–0.73	0.605	
	Subtree	1.53	1.45–1.62	0.08	−0.05–0.20	0.613	
Mass ∼ Diameter	Segment	2.09	2.01–2.17	−0.75	−0.87–0.64	0.825	
	Path	2.33	2.27–2.40	−0.90	−0.99–0.81	0.920	
	Subtree	2.49	2.43–2.55	−1.05	−1.14–0.96	0.926	
(b) Apple	
Length ∼ Diameter	Segment	−1.15	−1.27–1.04	3.63	3.45–3.82	0.024	
	Path	1.10	1.02–1.19	0.32	0.19–0.46	0.442	
	Subtree	1.65	1.54–1.76	−0.43	−0.61–0.26	0.577	
Mass ∼ Diameter	Segment	2.11	1.99–2.23	−0.67	−0.87–0.48	0.688	
	Path	2.36	2.28–2.45	−0.93	−1.07–0.79	0.867	
 	Subtree	2.57	2.49–2.66	−1.21	−1.35–1.07	0.892	

Figure 2 Estimates of allometric exponent (a) and 95% confidence intervals for five 24-year-old tart cherry and nineteen 10-year-old ‘Golden Delicious’ apple for each branch-level classification.

Segment level estimates are marked by triangles, path by diamonds, and subtree by squares with tart cherry shaded and apple open symbol. The predicted a from the process-bases models are marked as horizontal lines. (A–F) The predicted a from the FS model: elastic similarity is marked by a dashed line and flow similarity by a dot-dash line. (G–H) The predicted a from the WBE model is marked by a dashed line.

Branch classification

Across all morphological characteristics, allometric relationships at the path and subtree levels tended to have equal or higher r2 values relative to the same relationship calculated at the segment-level (Table 2, Fig. 2). Allometric relationships of multi-dimensional morphological characteristics, like surface area and volume, tend to exhibit higher r2 values than allometric relationships using linear morphological characteristics, like length and diameter (Fig. 2).

Model expectations

When compared to the predictions of the FS model, five of six species-level a estimations at the path level and four of six at the subtree level fall within the expected ranges described by flow similarity and elastic similarity constraints (Fig. 2). The relationship that does not fall within the expected range at the path level is diameter ∼ volume (Fig. 2C), while the two relationships that do not fall within the expected range at the subtree level are length ∼ volume (Fig. 2D) and length ∼ surface area (Fig. 2F). Only one set of species-level a estimations at the segment level (length ∼ volume) narrowly falls within the expected range of FS for both species (Fig. 2D). Estimated a tend to shift among segment, path and subtree level branch classification in the direction towards the flow similarity expectation of FS. At a species level, the path and subtree level mass ∼ diameter relationships express a very strong relationship with an estimated a close to the expected value from WBE (Fig. 2H). The subtree level relationship confidence intervals overlap with the empirical estimation of 2.53 by Brown (1997; Table 2). The length ∼ mass relationship estimated a at a species level appears to deviate significantly from the WBE prediction (Fig. 2G).

Discussion

Our study describes the allometry of two variably managed orchard tree species drawing context from existing process-based allometry models (Table 2). The allometric relationships of the sampled orchard trees are broadly consistent with each other and the expectations of the process-based allometry models. We find overlap in 95% confidence intervals of estimated a for tart cherry and apple for most of the allometric relationships evaluated at the path-level (6/9) and subtree-level (7/9) indicating a consistent pattern in growth and resulting tree architecture (Fig. 2). The allometric relationships evaluated for tart cherry and apple individuals sampled are well described by a power law and the empirical estimates of a tend to fall within the bounds of the expectations of FS and overlap with WBE particularly for the relationships defined by diameter and mass. Management of these orchard trees does appear to have some effect on the allometric relationships in the manipulation of the length dimension and the relaxation of biomechanical constraints (Shinozaki et al., 1964). Estimates of a shift significantly for both species between segment, path and subtree branch classifications blurring the scale at which these allometries are consistent within individuals.

Strong allometric relationships in orchard trees provide support for similar patterns in plant growth, despite different genetic material and management approaches driving growth and architecture among cherry and apple systems. Allometric relationships with multi-dimensional branch dimensions (i.e., surface area and volume) are stronger than relationships with linear dimensions (i.e., length and diameter; Fig. 2). This could be because these multi-dimensional branch dimensions better reflect resource transport and environmental exposure (Price et al., 2015; West, Brown & Enquist, 1997). For instance, surface area might relate to the number of leaves distributed on a branch, dictating the photosynthetic capacity of that location on the plant (Allen, Prusinkiewicz & DeJong, 2005). The volume of a branch might be a better predictor of water use than either length or diameter independently (McCulloh & Sperry, 2005). Better performance of multi-dimensional parameters linked to environmental exposure could explain why we find stronger morphological relationships also emerge at the path and subtree level; though, many of the strongest allometric relationships are represented by the highest r2 values for all branch classifications (Figs. 2B, 2C, 2H, 2I).

Existing process-based allometric models, WBE and FS, derive expectations of a for idealized plants. These models provide context for understanding how physiological mechanisms drive the conservation or deviation of plant growth from the idealized expectations (Table 1). The mass ∼ diameter relationship is most consistent among species and individuals within our study and is in general agreement with the expected a of the WBE, despite genetic and management differences (Table 2, Fig. 2H). The diameter ∼ volume and diameter ∼ area also tend to confer the elastic similarity expectation (Figs. 2C 2E). The other allometric relationships explored including length dimensions are consistently described by estimated a that shift within the expectations of idealized plants towards flow similarity and away from the biomechanical constraint of elastic similarity (Fig. 2).

The shift in a towards flow similarity for allometries including length dimensions may be attributed to the genetic selection for improved production efficiency or to substantial manipulation of tree architecture for commercial fruit production. This study represents two distinct species with clonal genetic material developed through selective breeding. The cherries are clonal scion material with closely related seedling rootstocks, while the apples are clonal scion with five distinct clonal rootstocks that were selected for known differences in tree growth, architecture, and production efficiency (Marini et al., 2014). Manipulation of tree architecture through pruning directly influences length and length-including branch dimensions (i.e., surface area, volume). As much as 25% of total aboveground biomass is removed annually, which reduces the growth potential of a tree by reducing leaf area and altering the root:shoot balance (Ferree & Schupp, 2003). Localized effects of pruning changes the load-bearing status and growth potential in basal branches. Remaining branches are mainly structurally important branches with wide crotch angles and relatively stubby dimensions suited for bearing a commercial fruit load (Ferree & Schupp, 2003). Regrowth is invigorated with increased investment in nearby fruit-bearing spurs and lateral branches, potentially driving the shift in a towards flow similarity (Grochowska et al., 1984; Fumey et al., 2011). In addition, training of tree architecture with structural supports may influence branch dimensions and localized constraints on branch physiology. Though not statistically significant, we observe that apple is described by an a that shifts slightly more towards flow similarity in relation to tart cherry. The greater shift towards flow similarity in apple is consistent with a relaxation of biomechanical constraints due to direct structural support provided by trellises in the apple system and more intensive ‘length-reducing’ pruning cuts. In contrast, the tart cherries are free standing and receive minimal ‘branch-removing’ pruning cuts that may be less of a factor in relaxing biomechanical constraints.

The biological process behind strong and consistent allometric relationships is linked to physiological limitations of plant growth in unmanaged plants (McMahon & Kronauer, 1976; Niklas & Spatz, 2004; Savage et al., 2010). It has remained difficult to disentangle the limiting effects of biomechanical and hydraulic processes, but the insights of the processed-based allometry models provide the opportunity for a first attempt at exploring these constraints in domesticated plants. From this study, it appears that both biomechanical and hydraulic constraints are limiting plant function in orchard systems. Biomechanical constraints may define the diameter of branches while branch length may respond to a relaxation of biomechanical constraints to be hydraulically limited (Fig. 2). Biomechanical and hydraulic constraints may also be acting at different levels of branch classification as represented by the shift in estimated a among scales. Individual segments are more constrained by load-bearing than what is observed at the path and subtree levels demonstrated by the segment level estimated a shift towards the biomechanically driven elastic similarity expectation while the path and subtree levels estimated a generally shift towards the hydraulically driven flow similarity expectation (Fig. 2). This within-canopy shift from elastic similarity to flow similarity may support the concept of ‘incomplete branch autonomy’ by which branches organize themselves independently due to the localized distribution of leaves and the resulting photosynthetic material but ultimately interact within a tree-wide network of nutrient transport and hormone signaling pathways (Marsal et al., 2003). The development of theory that disentangles biomechanical and hydraulic constraints provides the opportunity for further exploration of these physiological mechanisms as they relate to plant allometry.

Continued use of a process-based allometric approach in orchard systems may lead to a more general understanding of plant growth that can be linked to physiology and, in the case of orchard trees, can inform management techniques and research programs designed to maintain plant health, increase yields, and reduce resource use (Costes, 2004; Niklas, 2004; Lauri & Claverie, 2008). Describing common allometric relationships and physiological limitations for orchard systems can reveal a boundary where constraints from physics drive plant function more than human intervention. This study finds that allometric relationships are largely invariant between the two orchard species tested, but that the estimated a vary within the idealized expectations of process-based allometric models likely due to the extreme human influence on the plants studied. Both tart cherries and apples were heavily pruned at some stage of their development and received fertilizer, water, and agrochemical applications at levels optimal for reproductive growth. We found that the architecture of the two orchard species are described by allometries indicative of plant growth with reduced biomechanical demands, consistent with the large removal of biomass from the plant during pruning. Despite the large removal of biomass for each of these species, growth following management appears compensatory in such a way that the mass ∼ diameter relationship returns to consistent and expected relationships, while length ∼ diameter relationships fluctuate potentially according to pruning intensity. Future research might focus on the facets of management that drive allometry the farthest from model expectations or use measurements of body size to standardize treatments that could provide improved analysis of competing orchard systems and varieties. Breeding programs might better identify varieties that are approaching the yield ceiling while optimizing for resource use efficiency and yield improvements in the varieties with greatest potential for improvement.

Conclusions

Our study finds strong allometric relationships in two variably managed orchard species that are broadly consistent among the species. Two process-based models provide context for understanding the potential effects of growing conditions and management on growth and physiology of orchard trees. Agreement with idealized model expectations is limited to the mass ∼ diameter relationship of WBE though the empirical estimates of allometric relationships tend to fall within the bounds of the FS towards the flow similarity expectation. This study reveals the potential for continued use of process-based allometry within agricultural systems; however, expectations derived for idealized plants may be insufficient alone for the description of orchard systems due the human manipulation of plants’ physiology and growing conditions. Although orchard trees are particularly complex candidates due to genetic, environmental and physical manipulation, process-based allometry may still provide a mechanistic understanding of the effects of management for optimal reproductive growth.

This paper was greatly improved through discussion and review by C Price, B Black, K Mott, J Reeve, and E White. Orchard blocks sampled were maintained by T Lindstrom at the Kaysville Research Farm, Kaysville, UT. Assistance with field sampling was provided by J Anderson, J Shugart, and multiple members of the Weecology lab.

Additional Information and Declarations

Competing Interests

Author Contributions

Data Availability

The authors declare there are no competing interests.

Zachary T. Brym conceived and designed the experiments, performed the experiments, analyzed the data, prepared figures and/or tables, authored or reviewed drafts of the paper, approved the final draft.

S.K. Morgan Ernest conceived and designed the experiments, authored or reviewed drafts of the paper, approved the final draft.

The following information was supplied regarding data availability:

GitHub: https://github.com/weecology/branch-arch/tree/master/GeneralAllometry.

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
