# Peer review of "Process-based allometry describes the influence of management on orchard tree aboveground architecture"

_PeerJ, doi:10.7717/peerj.4949_

## Round 0.1 · original submission · Major Revisions

I now received two reviews and both of them are relatively positive. The two reviewers provide some very constructive comments and suggestions, and I think you should adopt them. Reviewer #1 pointed out the issue of the citation of Price et al. arXiv, which seems pretty major to me. This is not a refereed paper which may weaken the soundness of this piece. You should find published papers to support the flow similarity model.

Reviewer 1 ·

Basic reporting

The paper is well written, makes good use of citations (though I have problems with one in particular and would like to see some additional ones). The structure is fine, raw data is available.

The manuscript is generally self-consistent, though some papers I recommend citing change at least one conclusion.

Experimental design

No comment

Validity of the findings

Data is robust and freely available, so others can replicate the statistical analysis.

In general "Flow Similarity" is a good addition to scaling theory, but I have problems with the fact that a major paper cited in this manuscript has not been peer reviewed. That weakens the overall conclusions.

Additional comments

Paper title: Process-based allometry describes the influence of management on orchard tree aboveground architecture

Using two orchard tress species (cherry and apple) the authors examined inter- and intraspecific scaling relationships, comparing results to those predicted by the WBE model and the "Flow Similarity" model.

General comments: The paper is clearly written and addresses a question I have thought about myself--do the scaling relationships of managed crop trees that are pruned differ from, say, wild trees or trees grown in a plantation setting. In general I enjoyed the paper and find their methods and interpretation of results sound. I have some specific concerns, however, which fall into a few categories:
1.) materials cited
2.) possible conflation between WBE idea of branching and flow similarity's concept of branching
3.) way data is presented

See below for details.

Specific comments:

The Flow Similarity model builds on actual branching where as the WBE concept of branching is more abstract and is more geared at less-macro scales. I am slightly concerned you are conflating the two branching ideas. At teh very least it would be nice to discuss the differences (and more importantly the similarities) between the two, and why they can be treated as equivilent in this work. Maybe cite Duncan 2013 "Deviation from symmetrically self-similar branching in trees predicts altered hydraulics, mechanics, light interception, and metabolic scaling."

Line 41: The citation of Huxley as 1972 is misleading. "Problems of Relative Growth" was initially published in 1932, and predates the joint paper by Huxley and Tessier where they unified the language used (allometry/isometry) and established the accepted equation form y=bx^a. Consider using the 1932 publication year or cite the 1936 paper that first introduced the word "allometry" into the literature.
Huxley, Julian S, and Georges Teissier. 1936. “Terminology of Relative Growth.” Nature 137 (3471): 780–81.

Line 46: In keeping with talking about Huxley and the 1936 paper, the equation on line 46 different from the accepted standard format established by Huxley. I recommend y=bx^a vs using y0

Line 48: The term "fitted parameters" is slightly misleading in my mind. In log-log space, alpha is the slope of the line and beta is the Y intercept. This description (I think) makes the allometric equation more accessable.

Lines 50-51, 68-69: "Because many of these models predict only a single value of a for any particular x-y relationship, the exact value an importance of a is much debated." Enquist at al 2007 "Biological Scaling: Does the exception prove the rule" discusses an expansion of the WBE and specifically shows a plot of branch length vs branch diameter for seedlings and small plants( a=~1.8), and small to large trees (a=0.655). That the WBE allows for a range of a depending on the developmental stage of the tree contradicts your assertion that WBE predits " only a single value of a for any particular x-y relationship." It should also be noted that the values provided in Enquist 2007 match the FS predicted changing exponent values for L vs D (2 to 2/3)(Table 1)

Line 51: "...the exact value an importance of a is much debated." This debate goes all the way to Huxley's publication of "Problems of Relative Growth" in 1932. D'Arcy Thompson was critical of Huxley's allometric equations saying "Julian Huxley holds, and many hold with him, that the exponential or logarithmic formula, or compound-interest law, is of general application to cases of differential growth-rates. I do not find it to be so: any more than we have found organ, organism or population to increase by compound interest or geometrical progression, save under exceptional circumstances and in transient phase. Undoubtedly many of Huxley's instances shew increase by compound interest, during a phase of rapid and unstinted growth; but I find many others following a simple-interest rather than compound-interest law." (Thompson 1942. "On Growth and Form"). You could also cite any number of modern papers by Packard that present arguments similar to Thompson's (Packard 2014. "Assessing Allometric Growth by Leaves and the Hypothesis of Diminishing Returns")

Lines 53-55: The following sentence could use a reference to back it up: "Determining whether empirically estimated a support an expected value is challenging because while the values of exponents tenS to be constraineS, there is still variation within and across species." Citing Niklas 1999 "Plant allometry: is there a grand unifying theory" would work, though you might already cite a paper that covers it.

Line 58: The Price paper is problematic in my eyes. The paper you list as a preprint was uploaded to arXiv in 2015. Papers submitted to arXiv are not peer reviewed. I am at all opposed to papers on arXiv. I simply bring it up because the flow similarity model is a major component in the paper, and as far as I know there has never been a peer-reviewed follow up to this preprint. Your paper heavily relies on material in a paper that hasn't undergone the peer-review process.

Line 60, 83-84, 96-98: "area-preserving branching," archetecture manipulated by pruning, how trees respond to human manipulation: There is a very long and historically relevant history around this idea. Leonardo da Vinci is the first person known to propose what would eventually become known as the “pipe model”. In his backward writing he noted:

“All the branches of a tree at every stage of its height when put together are equal in thickness to the trunk [below them]. All the branches of a water [course] at every stage of its course, if they are of equal rapidity, are equal to the body of the main stream.

“Every year when the boughs of a plant have made an end of maturing their growth, they will have made, when put together, a thickness equal to that of the main stem; and at every stage of its ramification you will find the thickness of the said main stem as i k, g h, e f, c d, a b, will always be equal to each other, unless the tree is pollard--if so the rule does not hold good.” (see attached image for the sketch the letters refer to. Remember that he wrote backward, so flipping the image will correct the labeling)

It strikes me that his passage is directly applicable to this paper because the authors are looking at pruned orchard trees. Pollarded trees are an extreme form of pruning, but a form of pruning never-the-less. Leonardo made the observation about water flow in the same section in which he discussed tree branches, hinting that he sensed some relationship there. With the authors looking at both pruning and discussing hydraulic flow, it seems like a perfect quotation to at least mention.

The author’s assertion that “Management of these orchard trees does appear to have some effect on the allometric relationships in the manipulation of the length dimension and the relaxation of biomechanics constraints” confirms da Vinci’s anecdotal observation. The authors and journal might be able to use that to promote the paper in popular press outlets.

The more modern citation for the pipe-model is:
Shinozaki K, Yoda K, Hozumi K, Kira T. 1964. A quantitative analysis of plant form–the Pipe Model Theory: I. Basic analysis. Japanese Journal of Ecology 14: 97–105.

Lines 178-181: I definitely appreciate your acknowledgement that there is a vocal group of researchers that question the use of log-log plots when examining data. For some background you could cite the "Differing Persepctives" papers in 2014 175:7 "International Journal of Plant Sciences": Niklas and Hammond "Assessing Scaling Relationships: Uses, Abuses, and Alternatives" and Packard "Assessing Allometric Growth by Leaves and the Hypothesis of Diminishing Returns"

Lines 212-213: Is listing the license in the paper necessary? The authors should list the license on the GitHub page. I very much approve of the authors providing the data and code used in the analysis, though.

Lines 185-187: I very much appreciate that the raw data and tools for analysing the data are available on github.

Line 251: "idealized plants from physics first principles" is awkwardly phrased.

Figure 1: This is a VERY minor thing, but figure 1 could be more attractive if rendered using an L-system/context free graphical package.

Figure 2: I again applaud the author’s presentation of data in a graphical fashion, but I’m not sure presenting predicted exponent values, 95% confidence values, and r2 as a series of figures is necessary. From a practical perspective, tables are cheaper to publish than tables. From a reviewer’s perspective, presenting this data in a graphical format does improve my understanding over the same data in a table.

Figure 2, Supplemental S1: to clarify, do you mean R^2 or r^2? In the paper you use "r^2" which referrs to simple linear regression, but the figures use R2.

Supplemental Figure S1: Is it necessary to include this? Also, does it make the most sense to convey r2 values as a figure? While it graphically shows qualitative differences, it is difficult to get quantitative values when the data is presented in this way. But, again, is Figure 1S necessary?

Supplemental Figure S2: Similar comment about how the data is conveyed as said for Fig S1. I have not encountered allometric exponent expectations, measurements, 95% confidence intervals, and r^2(or R2^2?) portrayed like this before. That is not a bad thing: I think it is important to think about how to present data in meaningful ways. But, for example, circle size indicating r^2 doesn't tell me _values_ for the r^2. I can make estimates in certain ranges, but am less secure in assigning actual values. Can Supplemental Figure 2 (and all it’s sub figures) be presented as a table?

Supplemental Figure S3: It seems like the authors plotted everything against everything. I am not convinced all these plots are necessary.

Reviewer 2 ·

Basic reporting

no comment

Experimental design

no comment

Validity of the findings

no comment

Additional comments

This is a nice paper that compares empirical allometry data for branches of two orchard trees to predictions from two mechanistic plant allometry models. The authors find that data generally agree with predictions of the models, and for cases where data do not, authors discuss why orchard management practices may cause these deviations. The paper is very well written and the research questions are clear and well-defined. The discussion of why orchard trees deviate in some aspects from theory predictions is interesting. I also enjoyed the discussion of how allometry and process-based models can inform management techniques. The data and analyses appear to have been conducted properly. I have only a couple of minor comments for the authors.

First, support for FS is interpreted as falling within range of predictions for elastic similarity and flow similarity. This is a weak test however since the ranges are generally large (Table 1). Nevertheless, the fact that two relationships do not fall within this range might be interpreted as stronger evidence that FS doesn’t control these relationships in orchard trees. These issues could be noted in the text.

Second, Fig. S1, S2 – I’m confused by the caption. Are these just plots of the r2 for various fits? How does this relate to “strongest model fitted?” Why does caption mention AICc, is that shown in Figure? Perhaps a different symbol or color could be used to identify best fitting model.

---

## Round 0.2 · accepted · Accept

I have received reviews from the original 2 reviewers, and the feedbacks are all positive. Good job!

# Reviewer 1 ·

Basic reporting

My previous review of this paper noted that the submission was well written, made good use of citations, was structured in a reasonable way, and made the raw data available for validation.

The authors have only improved on their previous draft.

Experimental design

No comment

Validity of the findings

As stated previously: data is robust and freely available so others can replicate the statistical analysis.

My major concern--regarding the role of "flow similarity" in the absence of a peer reviewed paper--has been removed. I think your approach to presenting as a contextual piece is a good way to deal with the issue.

Additional comments

Follow-up review of the paper "Process-based allometry describes the influence of management on orchard tree aboveground architecture."

It's gratifying to see that the authors chose to incorporate so many of the references I had suggested--in spite of my getting some names and years wrong. That's what happens when you try and site things from memory instead of looking things up. I was especially pleased that they found the Equist 2007 to be useful. It's my hope they find the additional referneces helped their work.

Figure 2: r<sup>2</sup> instead of R2, please

Reviewer 2 ·

Basic reporting

no comments

Experimental design

no comments

Validity of the findings

no comments

Additional comments

Line 16: “processed-based” should be “process-based”